# Evaluating Parameter-Efficient Finetuning Approaches for Pre-trained Models on the Financial Domain

**Isabella Olariu**◇⋆, **Cedric Lothritz**⋆, **Jacques Klein**⋆,
**Tegawendé F. Bissyandé**⋆, **Siwen Guo**◇, **Shohreh Haddadan**◇

◇ Zortify S.A. | 9, Rue du Laboratoire, L-1911 Gare Luxembourg
⋆University of Luxembourg | 6, Rue Coudenhove-Kalergi, L-1359 Luxembourg
{isabella, siwen, shohreh}@zortify.com
{cedric.lothritz, jacques.klein, tegawende.bissyande}@uni.lu

## Abstract

Large-scale language models with millions, billions, or trillions of trainable parameters are becoming increasingly popular. However, they risk becoming rapidly over-parameterized and the adaptation cost of fully fine-tuning them increases significantly. Storing them becomes progressively impractical as it requires keeping a separate copy of all the fine-tuned weights for each task. By freezing all pre-trained weights during fine-tuning, parameter-efficient tuning approaches have become an appealing alternative to traditional fine-tuning. The performance of these approaches has been evaluated on common NLP tasks of the GLUE benchmark and shown to match full fine-tuning performance, however, their impact is less researched in domain-specific fields such as finance. This work compares the performance of a set of financial BERT-like models to their fully fine-tuned counterparts by leveraging different parameter-efficient tuning methods. We see that results are comparable to traditional fine-tuning while gaining in time and resource efficiency.

## 1 Introduction

Natural Language Processing (NLP) and the increasing popularity of large language models have redefined how computers understand and interpret human language. In particular, the financial domain, fundamentally intertwined with language and communication, benefits from NLP techniques to process complex financial documents. This domain deals with a vast array of unstructured textual data coming from sources such as news articles, financial reports, or investor opinions, each offering potentially valuable insights for processes such as risk assessment or decision-making. To this end, numerous efforts have been made to tailor large language models such as BERT (Devlin et al., 2019) or GPT-3 (Brown et al., 2020) to the financial domain, resulting in models such as FinBERT(Araci, 2019;

Yang et al., 2020; Liu et al., 2021), FLANG-BERT (Shah et al., 2022) or BloombergGPT (Wu et al., 2023). Despite their immense potential, these pre-trained models require fine-tuning to adapt their parameters to specific tasks, which becomes prohibitively expensive the larger the models are.

Recently, more parameter-efficient alternatives to full fine-tuning such as Low-Rank Adaptation (LoRA; Hu et al., 2021), Prefix-Tuning (Li and Liang, 2021) or Adapters (Houlsby et al., 2019; Pfeiffer et al., 2020, 2021) have been proposed to mitigate the vast adaptation cost of language models while maintaining full fine-tuning performance (Rücklé et al., 2021). These relatively new approaches share the same underlying principle of freezing the pre-trained weights of the base model, while fine-tuning only a set of newly introduced parameters, resulting in a substantial reduction in the total number of trainable parameters. While parameter-efficient tuning methods have been evaluated on generic benchmarks such as GLUE (Wang et al., 2018) and shown to reach results that are comparable to full fine-tuning, they are less researched in domain-specific fields such as finance.

In this paper, we evaluate the efficacy of different parameter-efficient fine-tuning approaches applied to BERT-like models in the financial domain. Our contributions are twofold: **(a)** we compare the performance of parameter-efficient tuning approaches on a range of financial tasks against their fully fine-tuned counterparts, and **(b)** show both their model size and training time benefits.

## 2 Experimental Setup

In this section, we list our research questions, describe the experimental framework we set up to address these questions and briefly explain the parameter-efficient fine-tuning approaches that we apply.

## 2.1 Research Questions

We investigate the following research questions:
**RQ1:** How do parameter-efficient fine-tuning approaches perform on financial tasks?
**RQ2:** What are the advantages or disadvantages in terms of model size and time efficiency?

## 2.2 Baselines

We examine three baseline models for comparison purposes: BERT-base as our "generic" baseline, as well as FinBERT and FLANG-BERT as baselines specialized for financial documents.
**BERT-base:** BERT-base (Devlin et al., 2019) is a bidirectional Transformer model with 12 layers, 768 hidden units per layer, 12 attention heads, and a total of 110 million parameters.
**FinBERT:** FinBERT (Yang et al., 2020) is a domain-specific variant of the BERT model for the financial field. It was pre-trained on text derived from corporate reports, earnings call transcripts, and analyst reports, and uses a domain-adapted financial vocabulary. There are several FinBERT models (Araci, 2019; Liu et al., 2021), we use the same variant as in Shah et al. (2022) for this study.
**FLANG-BERT:** FLANG-BERT is part of a suite of recently released financial language models (Shah et al., 2022). It is specialized through further training BERT within the financial domain and employing finance-specific pre-training objectives for enhanced understanding of domain-specific terms.

## 2.3 Parameter-efficient Fine-tuning Techniques

**Low-Rank Adaptation.** LoRA is a technique that inserts trainable rank decomposition matrices into each layer of a Transformer model. During fine-tuning, it freezes all the model's weights except those of the newly added low-rank matrices, a process that significantly reduces the number of trainable parameters and minimizes the memory footprint during model training (Hu et al., 2021).
**Adapter-tuning.** Adapter-tuning adds a set of new parameters to a Transformer in the form of new layers. The design of adapters varies as different configurations and placements are possible (Stickland and Murray, 2019; Pfeiffer et al., 2020). The adapter layer typically consists of a down-projection, an up-projection, and a non-linear activation function, usually inserted after the multi-head attention or the feed-forward layer (Houlsby

et al., 2019). During fine-tuning, only the weights associated with the adapter layers and the prediction head are trained. Although there exist numerous variations of adapter-based fine-tuning, we use the approach of Houlsby et al. (2019) for this study.

## 2.4 Downstream Tasks

To evaluate the performance of parameter-efficient tuning techniques, we fine-tune our pre-trained models using adapters and LoRA on four tasks from the Financial Language Understanding Evaluation (FLUE) benchmark (Shah et al., 2022).
**Sentiment Classification.** We use the Financial Phrasebank dataset (FPB; Malo et al. (2014)) to recognize sentiment from financial news. The dataset consists of 4845 sentences categorized into *positive*, *negative*, and *neutral* labels.
**Sentiment Regression.** We perform aspect-based sentiment analysis using the publicly available Financial Question Answering (FiQA) dataset (Maia et al., 2018), containing 1173 samples from headlines, news statements, and microblogs. Sentiment scores range from $-1$ to $1$.
**News Headline Classification.** To detect time-sensitive data in financial news headlines, such as price fluctuations, we use the Gold News Headlines dataset (Sinha and Khandait, 2020). It comprises 11 412 news headlines and nine corresponding tags such as *Future Price*, *Past News*, or *Asset Comparison*. The aim is to perform binary classification for all nine labels.
**Named Entity Recognition.** Named Entity Recognition (NER) aims to identify key financial entities to enhance the understanding of the relationship between them. The dataset for this task consists of 1466 samples (Salinas Alvarado et al., 2015) and four entity types: PERSON (PER), LOCATION (LOC), ORGANISATION (ORG), and MISCELLANEOUS (MISC).

## 2.5 Fine-tuning Parameters

We conduct hyperparameter searches using BERT-base for each task to determine the best values for the learning rate and batch size. The number of epochs varies across each fine-tuning approach, it ranges from 1 to 5 for full fine-tuning, 1 to 100 for LoRA, and 1 to 20 for Adapter$_H$. The learning rate is chosen from {1e-7, 1e-6, 1e-5, 2e-5, 3e-5, 5e-5, 1e-4, 1e-3} and the batch size from {8, 16, 32}. For LoRA, we further search for the

rank[1], alpha value[2], and dropout rate in $\{1, 8, 16\}$, $\{1, 8, 16, 32\}$, and $\{0.01, 0.05, 0.1\}$ respectively. We re-use the same parameters for FinBERT and FLANG-BERT as they share the same base architecture. See Appendix A.1 for details about the resulting hyperparameters used for each task.

## 3 Experimental Results

In this section, we present the results from our experiments across four downstream tasks and answer the research questions introduced in Section 2. For each task, we fine-tune the pre-trained models over five runs and report the average performance as the final metric. Results are shown in Table 1.

### 3.1 RQ1: How do parameter-efficient fine-tuning approaches perform on financial tasks?

Table 1 shows the results across four tasks of all models fine-tuned in a traditional way as well as fine-tuned with LoRA and the Houlsby adapter (Adapter$_H$) by Houlsby et al. (2019). BERT-base, FinBERT, and FLANG-BERT represent the performance results obtained by Shah et al. (2022) on the FLUE benchmark. We reproduced these results, but because our numbers differ slightly from the original ones, we also report them in Table 1 and mark them by "(repr.)". We compare LoRA and Adapter$_H$ against our reproductions. As shown in Table 1, applying LoRA and Adapter$_H$ leads to results that are comparable to the fully fine-tuned models for some of the tasks.

For FPB SC, BERT-base + Adapter$_H$ outperforms BERT-base + LoRA and performs as well as BERT-base. BERT-base + Adapter$_H$ also performs better than BERT-base and BERT-base + LoRA on FiQA SR, and outperforms BERT-base + LoRA on Headline while reaching the same performance as fully fine-tuned BERT-base.

FinBERT + LoRA outperforms the adapter-tuned and fully fine-tuned versions of the baseline on FPB SC and FiQA SR, respectively. FLANG-BERT + Adapter$_H$ beats FLANG-BERT + LoRA on all tasks except on NER, and outperforms the fully fine-tuned FLANG-BERT on FiQA SR.

None of the parameter-efficient approaches outperform the baselines on NER, but FinBERT with both LoRA and Adapter$_H$ closely reaches that baseline performance.

The advantage of one method over the other, is not definitive, as the effectiveness of each approach in terms of performance varies depending on the task. Although, it can be seen that Adapter$_H$ does not underperform the baseline for each model variant on all of the tasks except for NER. Overall, these results suggest that parameter-efficient approaches are a suitable alternative to full fine-tuning and do not significantly hurt performance on financial tasks.

### 3.2 RQ2: What are the advantages or disadvantages in terms of model size and time efficiency?

One clear advantage of LoRA and Adapter$_H$, is the significant reduction in model size, as for each task only the newly added parameters are stored in the model file. Table 2 shows the variations in size and time taken for fine-tuning on Headline Classification.[3] Overall, we can see that the LoRA approach is the most parameter-efficient variant with at most $0.3\%$ of trainable parameters for the Headline Classification task. We can see the same trend for the other tasks in Appendix A.4 where LoRA requires as little as $0.04\%$ of updatable parameters for FiQA SR and and FPB.

In terms of time efficiency, applying LoRA and Adapter$_H$ reduced the training time for the largest dataset used for Headline Classification. For other tasks, the gain might not be as evident as shown in Appendix A.4.

## 4 Discussion

Our experiments show that performance on financial tasks is not necessarily compromised with parameter-efficient tuning approaches. We see that even on the smaller FiQA and NER datasets for Fin-BERT, these approaches achieve almost the same performance as full fine-tuning. Due to the small set of trainable parameters, the models might be less at risk of overfitting and catastrophic forgetting (He et al., 2021).

Regarding the performance between the two parameter-efficient methods, we argue that Adapter$_H$ performs better than LoRA. It almost always maintains full fine-tuning performance, and tends to outperform LoRA more frequently. We believe this to be the case because, compared to LoRA, Adapter$_H$ keeps a bigger share of trainable parameters. In addition, LoRA needs a much larger

---

[1]Dimension of the low-rank matrices
[2]Scaling factor for the weight matrices

[3]See Appendix A.4 for the remaining tasks.

| Model | FPB SC | FiQA SR | Headline | NER |
|---|---|---|---|---|
| Metric | Accuracy | MSE | Mean F-1 | F-1 |
| BERT-base | 0.86 | 0.07 | 0.97 | 0.79 |
| BERT-base (repr.) | 0.86 | 0.12 | 0.97 | **0.81** |
| BERT-base + LoRA | 0.83 | 0.16 | 0.95 | 0.75 |
| BERT-base + Adapter$_H$ | **0.86** | **0.10** | **0.97** | 0.77 |
| FinBERT | 0.87 | 0.07 | 0.97 | 0.80 |
| FinBERT (repr.) | 0.86 | 0.09 | 0.97 | **0.77** |
| FinBERT + LoRA | **0.89** | **0.08** | 0.94 | 0.76 |
| FinBERT + Adapter$_H$ | 0.86 | 0.09 | **0.97** | 0.76 |
| FLANG-BERT | 0.91 | 0.05 | 0.97 | 0.83 |
| FLANG-BERT (repr.) | 0.86 | 0.08 | 0.97 | **0.81** |
| FLANG-BERT + LoRA | 0.84 | 0.17 | 0.95 | 0.78 |
| FLANG-BERT + Adapter$_H$ | **0.86** | **0.07** | **0.97** | 0.76 |

Table 1: Summary of results on downstream tasks. Mean F-1 is the average F-1 score taken across all nine labels of the News Headline dataset. Breakdown of results for each label can be found in Appendix A.3. Numbers marked in bold highlight the best-performing fine-tuning approach per model.

| Model | Size | Trainable Parameters | | Fine-tuning Time |
|---|---|---|---|---|
| BERT-base | 418.0 MB | 109 780 228 | (100%) | 187 sec |
| BERT-base + LoRA | 1.2 MB | 297 988 | (0.3%) | 152 sec |
| BERT-base + Adapter$_H$ | 3.5 MB | 1 486 658 | (1.4%) | 122 sec |
| FinBERT | 419.0 MB | 110 049 796 | (100%) | 307 sec |
| FinBERT + LoRA | 1.2 MB | 297 988 | (0.3%) | 147 sec |
| FinBERT + Adapter$_H$ | 3.5 MB | 1 486 658 | (1.4%) | 131 sec |
| FLANG-BERT | 418.0 MB | 109 780 228 | (100%) | 305 sec |
| FLANG-BERT + LoRA | 1.2 MB | 297 988 | (0.3%) | 145 sec |
| FLANG-BERT + Adapter$_H$ | 3.5 MB | 1 486 658 | (1.4%) | 117 sec |

Table 2: Comparison of model size, number of trainable parameters and time for Headline Classification.

number of epochs to be able to reach a comparable performance to full fine-tuning (Hu et al., 2021; Chen et al., 2022).

From a practical point of view, the gain in storage space is apparent and allows to keep and load numerous fine-tuned models for an abundance of tasks locally. From a time-saving perspective, the datasets used in this study are relatively small so that even the full fine-tuning requires only a few minutes in a lot of cases. Our experiments indicate that a larger gain in time efficiency is observed for parameter-efficient techniques when applied to larger models and bigger datasets.

## 5  Related Work

Numerous studies have been released on the effectiveness of parameter-efficient approaches on the GLUE benchmark (Pfeiffer et al., 2020; Rücklé et al., 2021; He et al., 2021). In the financial domain, a number of studies build language models

from scratch or through continual in-domain pre-training and compare the domain-specific models to general-domain baseline models (Peng et al., 2021; Yang et al., 2020; Shah et al., 2022), but do not employ parameter-efficient techniques. Large language models for finance are on the rise (Wu et al., 2023; Xie et al., 2023) but have either access restrictions or do not take advantage of parameter-efficient tuning approaches. To the best of our knowledge, the first financial language model, fine-tuned with LoRA in combination with reinforcement learning, was released (Yang et al., 2023).

## 6  Conclusion

In this paper, we evaluated the effectiveness of LoRA and adapter fine-tuning applied to tasks in the financial domain. We showed that in a financial setting, we can reach full fine-tuning performance with smaller model sizes which allows for easier storage and sharing of fine-tuned models among

practitioners.

We would like to aspire future research to evaluate more parameter-efficient strategies and extend them to other domain-specific fields. Based on our results and with the increasing number of larger language models in the financial sector (Wu et al., 2023; Lu et al., 2023), we recommend the implementation of more parameter-efficient fine-tuning practices to strive towards reducing their carbon footprint and environmental impact.

## 7 Limitations

Similar to the majority of experimental research, this study may encounter potential limitations and validity threats.

First, the outcomes of our experiments might depend on the volume of data used for the different downstream tasks. We did not experiment with distinct dataset sizes for the FPB SA and News Headline Classification tasks, where more training samples are available, to see the impact on performance if we reduced the number of samples.

Second, while there are other parameter-efficient fine-tuning methods, our study is restricted to the use of LoRA and Houlsby Adapter. We did not assess the performance of other techniques to show their potential on financial tasks. In addition, we did not experiment with different adapter hidden sizes that might have an impact on performance.

Lastly, our hyperparameters differ from the ones employed in the baseline models. We conducted a hyperparameter search for each task to obtain the best configurations, however, for some tasks the number of epochs used for the fine-tuning varies a lot between the fully fine-tuned models and their efficiently fine-tuned counterparts. Hence, some results could be attributed to confounding variables that we did not manage to regulate.

## 8 Ethical Considerations

For this study, we fine-tuned our models using different openly available financial text corpora included in the FLUE benchmark (Shah et al., 2022). Despite the financial nature of the datasets, none of them carry high ethical risks as the respective authors ensure that they do not reveal any sensitive data, nor contain any information that could allow to identify personal data.

## 9 Acknowledgements

This research was funded in part by the Luxembourgish National Fund (FNR), grant reference NCER22/IS/16570468/NCER-FT. The simulations were performed on the Luxembourg national supercomputer MeluXina. We gratefully acknowledge the LuxProvide teams for their expert support.

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

# A Appendix

## A.1 Hyperparameter Search Results

| Hyperparameters | #epochs | bs | lr | r | lora alpha | lora dropout |
|---|---|---|---|---|---|---|
| FT *FPB* | 5 | 32 | 2e-5 | - | - | - |
| FT *FiQA* | 5 | 16 | 1e-4 | - | - | - |
| FT *Headline* | 5 | 8 | 1e-2 | - | - | - |
| FT *NER* | 5 | 16 | 1e-4 | - | - | - |
| LoRA *FPB* | 80 | 16 | 2e-5 | 1 | 32 | 0.01 |
| LoRA *FiQA* | 80 | 16 | 2e-5 | 1 | 1 | 0.01 |
| LoRA *Headline* | 3 | 16 | 5e-5 | 8 | 16 | 0.05 |
| LoRA *NER* | 5 | 16 | 1e-3 | 16 | 16 | 0.05 |
| Adapter$_H$ *FPB* | 6 | 16 | 1e-4 | - | - | - |
| Adapter$_H$ *FiQA* | 6 | 16 | 1e-4 | - | - | - |
| Adapter$_H$ *Headline* | 6 | 16 | 1e-4 | - | - | - |
| Adapter$_H$ *NER* | 20 | 16 | 1e-4 | - | - | - |

Table 3: Hyperparameters used for full fine-tuning (FT), LoRA, and Adapter$_H$ on the downstream tasks.

## A.2 Description of Datasets

| Datasets | Train | Dev | Test | Classes |
|---|---|---|---|---|
| FPB (Malo et al., 2014) | 3488 | 388 | 969 | 3 |
| FiQA SA (Maia et al., 2018) | 822 | 117 | 234 | - |
| Headline (Sinha and Khandait, 2020) | 7989 | 1141 | 2282 | 2 |
| NER (Salinas Alvarado et al., 2015) | 932 | 232 | 302 | 5 |

Table 4: Descriptive Statistics of FLUE datasets.

## A.3 Breakdown of results for News Headline Classification

| Label | Bert + LoRA | FinBERT + LoRA | FLANG-BERT + LoRA |
|---|---|---|---|
| Price or Not | 0.93 | 0.93 | 0.94 |
| Price Up | 0.90 | 0.92 | 0.92 |
| Price Constant | 0.94 | 0.94 | 0.93 |
| Price Down | 0.93 | 0.94 | 0.92 |
| Past Price | 0.93 | 0.92 | 0.93 |
| Future Price | 0.99 | 0.98 | 0.97 |
| Past News | 0.93 | 0.93 | 0.94 |
| Future News | 0.99 | 0.99 | 0.99 |
| Asset Comparison | 0.98 | 0.96 | 0.98 |
| Mean F-1 Score | 0.95 | 0.94 | 0.95 |

Table 5: Breakdown of LoRA results for each label for News Headline Classification.

| Label | Bert + Adapter$_H$ | FinBERT + Adapter$_H$ | FLANG-BERT + Adapter$_H$ |
|---|---|---|---|
| Price or Not | 0.96 | 0.95 | 0.95 |
| Price Up | 0.94 | 0.94 | 0.95 |
| Price Constant | 0.98 | 0.98 | 0.98 |
| Price Down | 0.95 | 0.95 | 0.95 |
| Past Price | 0.95 | 0.94 | 0.95 |
| Future Price | 0.99 | 0.99 | 0.99 |
| Past News | 0.96 | 0.95 | 0.95 |
| Future News | 0.99 | 0.99 | 0.99 |
| Asset Comparison | 0.99 | 0.99 | 0.99 |
| Mean F-1 Score | 0.97 | 0.97 | 0.97 |

Table 6: Breakdown of Adapter$_H$ results for each label
for News Headline Classification.

## A.4 Model Sizes and Time

### A.4.1 Financial Phrase Bank

| Model | Size | Trainable Parameters | Fine-tuning Time |
|---|---|---|---|
| BERT-base | 418 MB | 109 523 718 (100%) | 62 sec |
| BERT-base + LoRA | 171 KB | 41 478 (0.04%) | 1668 sec |
| BERT-base + Adapter$_H$ | 3.4 MB | 1 487 427 (1.4%) | 318 sec |
| FinBERT | 419 MB | 109 793 286 (100%) | 60 sec |
| FinBERT + LoRA | 1.2 MB | 41 478 (0.04%) | 1736 sec |
| FinBERT + Adapter$_H$ | 3.4 MB | 1 487 427 (1.4%) | 326 sec |
| FLANG-BERT | 418 MB | 109 523 718 (100%) | 60 sec |
| FLANG-BERT + LoRA | 171 KB | 41 478 (0.04%) | 1703 sec |
| FLANG-BERT + Adapter$_H$ | 3.4 MB | 1 487 427 (1.4%) | 326 sec |

Table 7: Comparison of model size, number of trainable
parameters and time for FPB.

### A.4.2 FiQA

| Model | Size | Trainable Parameters | Fine-tuning Time |
|---|---|---|---|
| BERT-base | 418 MB | 109 520 642 (100%) | 27 sec |
| BERT-base + LoRA | 165 KB | 38 402 (0.04%) | 700 sec |
| BERT-base + Adapter$_H$ | 3.4 MB | 1 485 889 (1.4%) | 46 sec |
| FinBERT | 419 MB | 109 790 210 (100%) | 84 sec |
| FinBERT + LoRA | 171 KB | 38 402 (0.04%) | 677 sec |
| FinBERT + Adapter$_H$ | 3.4 MB | 1 485 889 (1.4%) | 47 sec |
| FLANG-BERT | 418 MB | 109 520 642 (100%) | 35 sec |
| FLANG-BERT + LoRA | 165 KB | 38 402 (0.04%) | 649 sec |
| FLANG-BERT + Adapter$_H$ | 3.4 MB | 1 485 889 (1.4%) | 46 sec |

Table 8: Comparison of model size, number of trainable
parameters and time for FiQA.

### A.4.3 NER

| Model | Size | Trainable Parameters | Fine-tuning Time |
|---|---|---|---|
| BERT-base | 418 MB | 109 489 162 (100%) | 464 sec |
| BERT-base + LoRA | 2.7 MB | 699 658 (0.6%) | 330 sec |
| BERT-base + Adapter$_H$ | 3.4 MB | 898 373 (0.8%) | 1117 sec |
| FinBERT | 419 MB | 109 758 730 (100%) | 410 sec |
| FinBERT + LoRA | 2.7 MB | 699 658 (0.6%) | 331 sec |
| FinBERT + Adapter$_H$ | 3.4 MB | 898 373 (0.8%) | 1118 sec |
| FLANG-BERT | 418 MB | 109 489 162 (100%) | 429 sec |
| FLANG-BERT + LoRA | 2.7 MB | 699 658 (0.6%) | 330 sec |
| FLANG-BERT + Adapter$_H$ | 3.4 MB | 898 373 (0.8%) | 1118 sec |

Table 9: Comparison of model size, number of trainable parameters and time for NER.