# OpenReview forum: "Evaluating Parameter-Efficient Finetuning Approaches for Pre-trained Models on the Financial Domain"
_EMNLP/2023/Conference — EMNLP 2023 Findings_

### Official Review · Reviewer_JNPW · 2023-07-30

**Soundness:** 2

**Excitement:**

2: Mediocre: This paper makes marginal contributions (vs non-contemporaneous work), so I would rather not see it in the conference.

**Missing References:**

[1] He, J., Zhou, C., Ma, X., Berg-Kirkpatrick, T., & Neubig, G. (2021). Towards a unified view of parameter-efficient transfer learning.

[2] Liu, H., Tam, D., Muqeeth, M., Mohta, J., Huang, T., Bansal, M., & Raffel, C. A. (2022). Few-shot parameter-efficient fine-tuning is better and cheaper than in-context learning.


**Paper Topic And Main Contributions:**

This paper presents a study of PEFT in financial domain tasks. It addresses two research questions: 1. how do PEFTs perform on financial tasks? 2. what are the pros and cons of using PEFTs? The author conducts adapter-tuning and LoRA-tuning in comparison to fully supervised fine-tuning across BERT-base, FinBERT, and FLANG-BERT on 4 financial domain tasks from the FLUE benchmarks. However, the first research question is not well motivated as PEFTs have shown tremendous success across domains, languages, and even modalities. Therefore, the technical contribution of this paper is minimal, and there is no additional finding on PEFT on top of existing PEFT literature. Moreover, the second research question has been addressed by individual PEFT papers.

Pfeiffer, J., Goyal, N., Lin, X. V., Li, X., Cross, J., Riedel, S., & Artetxe, M. (2022). Lifting the Curse of Multilinguality by Pre-training Modular Transformers.

Chronopoulou, A., Peters, M. E., & Dodge, J. (2022). Efficient Hierarchical Domain Adaptation for Pretrained Language Models.
Sung, Y.-L., Cho, J., & Bansal, M. (2022). VL-Adapter: Parameter-Efficient Transfer Learning for Vision-and-Language Tasks.




**Questions For The Authors:**

1. Can you extend this analysis to more PEFT approaches and types of base models?

**Reasons To Accept:**

1. This paper extends the existing PEFT analysis to the financial domain, and indicates competitive performance to fully supervised fine-tuning.


**Reasons To Reject:**

1. The study of PEFT in a new domain is not motivated and has limited novelty.

2. The study only contains two PEFT approaches, Adapter and LoRA. More interesting comparisons among parallel adapter [1], IA^3, pre-fix tuning, prompt tuning, and many other new PEFT approaches can make the analysis in the finance domain more existing.

3. The study only focuses on classification-based finance tasks. Generative tasks can lead to more interesting results. Additionally, this study only considers encoder-only models. A more throughout evaluation study of PEFT should consider T5, and GPT-type models.

4. The second research question is not novel.


**Reproducibility:**

4: Could mostly reproduce the results, but there may be some variation because of sample variance or minor variations in their interpretation of the protocol or method.

**Reviewer Confidence:**

5: Positive that my evaluation is correct. I read the paper very carefully and I am very familiar with related work.

**Typos Grammar Style And Presentation Improvements:**

Line 03: they risk -> the risk / their risk
The bold fonts in Table 1 are misleading. The author should not exclude the original results in ranking and should also highlight the FT baseline that achieves the same highest score.

---

> ### Author Rebuttal · Authors · 2023-08-28
>
> **Response to Comment 1:**
> We appreciate the reviewer’s feedback regarding the motivation and novelty of our study. For the camera-ready version, we will emphasize on the significance of exploring PEFT techniques beyond the general context and highlight the potential benefits of these approaches in domain-specific contexts.
>
> **Response to Comment 2:**
> We thank the reviewer for the valuable suggestion to broaden the scope of our comparisons among PEFT approaches. We agree that including a wider range of methods would enrich the analysis and contribute to a more comprehensive understanding of PEFT approaches in the finance domain. To provide a more detailed and diverse comparison, we will incorporate these additional methods into our future work.
>
> **Response to Comment 3:**
> We acknowledge the reviewer's insight regarding the potential benefits of exploring generative tasks and considering a more diverse range of model types in our PEFT evaluation. To address this, we will extend our study in the future to also incorporate generative tasks. Furthermore, to make the analysis of PEFT more comprehensive, we will include an evaluation of PEFT on more model types.
>
> **Response to Comment 4:**
> We appreciate the reviewer's feedback regarding the novelty of our second research question. We believed the second research question to be relevant and to logically follow the conclusions of the first research question.
>
> **Response to Question 1:**
> We thank the reviewer for suggesting an extension of our analysis to encompass a broader range of PEFT approaches and base models. Building upon this suggestion, we plan to conduct an in-depth evaluation of additional PEFT techniques and explore their performance across different types of base models in future work.

---

### Official Review · Reviewer_yiyh · 2023-08-04

**Soundness:** 2

**Excitement:**

3: Ambivalent: It has merits (e.g., it reports state-of-the-art results, the idea is nice), but there are key weaknesses (e.g., it describes incremental work), and it can significantly benefit from another round of revision. However, I won't object to accepting it if my co-reviewers champion it.

**Paper Topic And Main Contributions:**

The authors report the performance of applying parameter-efficient methods, Adapters (Houlsby et al., 2019) and LoRA (Hu et al., 2021), to three backbone models: BERT-base, FinBERT, and FLANG-BERT on four finance data sets: FPB SC, FiAQ SR, Headline, and NER. The contribution of this work includes providing a preliminary study on parameter-efficient methods (i.e., $\text{Adapter}_H$ and LoRA) applied in the finance domain. The report concludes that the parameter-efficient method decreases the domain-specific BERT's (i.e., FinBERT's and FLANG-BERT) fine-tuning time by \~50% compared to the full fine-tuning (Table 2) for a relatively large "Headline" dataset (\~10K train+val). In addition, the performance is task-specific and might drop after incorporating parameter-efficient methods (Table 1, the first row of each model and lines +LORA and +$\text{Adapter}_H$).

**Questions For The Authors:**

- Question A: In Table 1, the reproduced results and the originally reported values (i.e., the second row with (repr.) and first row) are different. What is the reason for the performance gap? Please refer to the results reported in Table 2 of this work (Raj et al. 2022, https://aclanthology.org/2022.emnlp-main.148/). Are there reproducibility issues or a need for implementation details in the public code?
- Question B: Table 2 presents the fine-tuning time to decrease the Fine-tuning time. However, the same phenomenon only appears in Headline Classification datasets. Therefore, a follow-up question RQ2 is, for "smaller" datasets (Appendix A.4), what is the reason (+LoRA, or +$\text{Adapter}_H$) takes much more time for fine-tuning? For example, in Table 7, FinBERT + LoRA takes 1,736 sec, which is >10 times longer than the Baseline dataset shown in Table 2. Does setting $r$ to 1 make it hard to converge? What is the possible weakness of the parameter-efficient method in this scenario? They might be essential information for readers.
- Question C: Will the source code be public if accepted?

**Reasons To Accept:**

- The authors provide a brief report on applying the parameter-efficient method in the finance domain. The experiments show the task-specific manner of applying LORA to different size datasets. The increasing fine-tuning time of *BERT + LORA for smaller datasets shown in A.4 is an exciting topic for the community to explore.

**Reasons To Reject:**

- The correctness of the reproduced results in Table 1 needs further confirmation. The reproduced results and the originally reported values (i.e., the second row with (repr.) and the first row) are different with performance drop by up to 3% (see FinBERT/NER, 0.88 versus $\textbf{0.77}$), which is confusing in determining whether the parameter-efficient methods are feasible. For example, for FLANG-BERT, the original (i.e., FLANG-BERT) and reproduced MSE (i.e., FLANG-BERT (repr.)) in task "FiQA SR" are 0.05 and 0.08, respectively. If the MSE of FLANG-BERT+ $\text{Adapter}_H$ is 0.07, concluding "FLANG-BERT + $\text{Adapter}_H$ outperforms the fully fine-tuned FLANGBERT" in L208-L211 is not convincing for readers. More details for reproducing baseline models are required in L191-L194 to give more confidence in the correctness of Table 1.
- More conclusive results are required. For example, in section 3.1 L199-L215,  besides reporting the trend of the experiment results, the observation of why $\text{Adapter}_H$ looks more stable than LoRA is worth reporting.

**Reproducibility:**

4: Could mostly reproduce the results, but there may be some variation because of sample variance or minor variations in their interpretation of the protocol or method.

**Reviewer Confidence:**

4: Quite sure. I tried to check the important points carefully. It's unlikely, though conceivable, that I missed something that should affect my ratings.

**Typos Grammar Style And Presentation Improvements:**

- L211 typo: FLANGBERT -> FLANG-BERT

---

> ### Author Rebuttal · Authors · 2023-08-28
>
> **Response to Comment 1:**
> We appreciate the reviewer's attention to the correctness of the reproduced results in Table 1. We apologise for any confusion this may have caused. We would like to point out that we have stated to be using our reproduced results of the baselines due to some discrepancies with the originally reported values. We would further direct the reader to the Appendix for more results. We agree that the results are mixed and not conclusive enough to say that one method performs better than the other. We can only see a trend that Adapters seem to perform better than LoRA on the chosen tasks.
>
> **Response to Comment 2:**
> We thank the reviewer for highlighting the importance of having more conclusive results. In our revised manuscript, we will report on additional results between Adapter and LoRA. We would assume that because adapters keep a higher proportion of parameters compared to LoRA, that they seem more stable and will report further on this in the camera-ready version.
>
> **Response to Question A:**
> We thank the reviewer for this question and acknowledge the importance of addressing this discrepancy. We will provide an explanation of the potential reasons for the observed differences in the revised manuscript as well as reach out to the original authors to ask why there could be some potential differences in reproducing the results.
>
> **Response to Question B:**
> We thank the reviewer for raising questions about the differences in fine-tuning times observed across datasets and methods. As mentioned and seen in the literature, LoRA usually requires more epochs. We will make this clearer in the revised version of the paper.
>
> **Response to Question C:**
> We fully support the principles of transparency and reproducibility in research. As such, we are committed to making our source code publicly available if our paper is accepted. We will ensure to provide well-documented code that is accessible to other researchers who would like to replicate and further explore this topic.

---

### Official Review · Reviewer_7Xqg · 2023-08-10

**Soundness:** 1

**Excitement:**

2: Mediocre: This paper makes marginal contributions (vs non-contemporaneous work), so I would rather not see it in the conference.

**Paper Topic And Main Contributions:**


The study assesses the effectiveness of Parameter Efficient Fine-Tuning techniques on financial datasets. Specifically, the investigation delves into the performance of LoRA and Adapter-H methods when applied to Bert-base, FinBERT, and FlangBERT models. The datasets encompass a variety of tasks: 1) Soft Contextual Data Augmentation (Classification), 2) Financial Question Answering (FiQA) (Regression), 3) Gold News Headlines (Classification), and 4) Named Entity Recognition (NER).

The primary assertion of the paper is to demonstrate that these fine-tuning approaches yield comparable performance to that achieved through complete model fine-tuning. Notably, these techniques necessitate only a fraction of the parameters to be fine-tuned, effectively mitigating the risk of overfitting.

**Questions For The Authors:**

1) The paper's appeal could be enhanced by conducting evaluations across a wider range of tasks and diverse dataset types.
2) An examination of the scenarios where one method outperforms the other during model fine-tuning would add value. Insights into the determining factors behind such decisions should be explored.
3) It would provide valuable insights to understand the rationale behind the selection of specific hyperparameters and epoch counts. This transparency would contribute to a deeper understanding of the experimental setup.

**Reasons To Accept:**

The evaluation they conducted showcases relatively persuasive findings, indicating that the PEFT approach yields outcomes akin to full fine-tuning, all while avoiding overfitting.

**Reasons To Reject:**

1) The paper lacks significant contributions in terms of introducing new datasets, algorithms, or novel discoveries.
2) Various models underwent training for differing epoch counts, and any performance discrepancies could be linked to this factor.
3) The evaluation, as highlighted in the paper, is confined to a narrow selection of small datasets. It would be advisable to assess the approach across a broader spectrum of datasets.
4) The paper primarily revolves around fine-tuning models for financial data, yet it doesn't introduce any substantial new insights. Hence, it might find better alignment within workshops specifically focused on Finance NLP.

**Reproducibility:**

3: Could reproduce the results with some difficulty. The settings of parameters are underspecified or subjectively determined; the training/evaluation data are not widely available.

**Reviewer Confidence:**

5: Positive that my evaluation is correct. I read the paper very carefully and I am very familiar with related work.

---

> ### Author Rebuttal · Authors · 2023-08-28
>
> **Response to Comment 1:**
> We sincerely appreciate the reviewer's feedback and acknowledge the importance of making substantial contributions to the field. While we agree that introducing new datasets, algorithms, and novel discoveries is critical, we would like to emphasize that our primary objective was to evaluate the effectiveness of the LoRA and Adapter models in the context of fine-tuning for financial data. However, we take the reviewer's suggestion seriously and will explore opportunities to expand our work by incorporating additional datasets and different models  in future work.
>
> **Response to Comment 2:**
> We appreciate the reviewer pointing out the potential impact of varying training epoch counts on performance discrepancies. We strongly agree that this is a crucial factor to consider and ensure the validity of our findings. While conducting experiments with consistent hyperparameter settings, such as training epochs, is important, we chose the number of epochs based on a hyperparameter search that we conducted for every task. We found that the results reflect the general guidelines (https://docs.adapterhub.ml/training.html) of using a higher number of epochs for PEFT methods as seen in the LoRA literature (https://arxiv.org/pdf/2106.09685.pdf).
>
> **Response to Comment 3:**
> We appreciate the reviewer's suggestion to broaden the evaluation across a wider spectrum of datasets. We plan to extend our experiments to include a diverse range of datasets from various domains in order to better assess the effectiveness of PEFT techniques in future work.
>
> **Response to Comment 4:**
> We thank the reviewer for their insight regarding the potential alignment of our paper with workshops specifically focused on Finance NLP. We can consider a specialized workshop in the future.
>
> **Response to Question 1:**
> The reviewer’s suggestion to conduct evaluations across a wider range of tasks and dataset types is highly valued. We completely agree and are committed to expanding the scope of our experiments to encompass a diverse set of tasks and datasets in future work.
>
> **Response to Question 2:**
> We appreciate the reviewer's suggestion to investigate scenarios where one method outperforms the other during model fine-tuning. We will consider this more detailed analysis in future work in order to uncover the factors that contribute to the success of each method, determine the strengths and weaknesses of the approaches, and understand their suitability for different scenarios.
>
> **Response to Question 3:**
> We recognize the importance of transparency in hyperparameter selection to facilitate a deeper understanding of our experimental setup. We would like to point out that we mentioned in the paper that we did a hyperparameter search for each dataset and task, and will make this clearer in the revised version of the paper.

---

### Official Review · Reviewer_mQ9a · 2023-08-14

**Typos Grammar Style And Presentation Improvements:** No concerns here, paper was well-writ…
**Soundness:** 4

**Excitement:**

2: Mediocre: This paper makes marginal contributions (vs non-contemporaneous work), so I would rather not see it in the conference.

**Paper Topic And Main Contributions:**

This paper performs a thorough comparison of financial BERT-like models to their fully fine-tuned counterparts on financial tasks by leveraging efficient parameter-efficient tuning methods Adapters and LoRA.

**Questions For The Authors:**

I would have liked to see the paper address more interesting research questions such as a) how does the effectiveness of LoRA/Adapter change across datasets? b) Beyond vanilla Adapter/LoRA what else can one do to do better on new domains? c) Evaluating additional parameter-efficient methods and udnerstanding trends across datasets, domains, and tasks.

**Reasons To Accept:**

This paper shows that parameter-efficient finetuning methods like LoRA and Adapters are able to retain nearly all of the finetuning performance while gaining in time and resource efficiency.

**Reasons To Reject:**

1. Novelty: The paper's contribution to existing literature seems marginal. Evaluating well-established parameter efficient finetuning methods on new datasets (finance domain) is helpful but in my opinion, does not constitute a significant contribution to this conference.

**Reproducibility:**

4: Could mostly reproduce the results, but there may be some variation because of sample variance or minor variations in their interpretation of the protocol or method.

**Reviewer Confidence:**

4: Quite sure. I tried to check the important points carefully. It's unlikely, though conceivable, that I missed something that should affect my ratings.

---

> ### Author Rebuttal · Authors · 2023-08-28
>
> **Response to Comment 1:**
> We thank the reviewer for the remark about the paper's novelty. We agree that the contribution to existing literature might seem marginal, though believe still valuable to evaluate such established parameter-efficient finetuning approaches to unexplored domains to further consolidate their efficacy. We believed a short paper to be a good choice to address our research questions.
>
> **Response to Question a:**
> We greatly appreciate the reviewer’s feedback and insightful suggestions. Analysing the effectiveness of LoRA/Adapter across datasets would indeed enhance the comprehensiveness of our work and add valuable insights to the paper. We will go into a more detailed analysis of PEFT methods across datasets in the revised version of the paper.
>
> **Response to Question b:**
> We thank the reviewer for highlighting the importance of exploring strategies beyond LoRA and Adapters. Other domain adaptation strategies exist, such as unsupervised domain adaptation, that could be leveraged or even combined with parameter-efficient finetuning methods to do better on new domains. In future works we will aim to present a more comprehensive framework for improving model performance in new domains by exploring techniques such as cross-domain pre-training, transfer learning, as well as considering external sources such as knowledge graphs.
>
> **Response to Question c:**
> We appreciate the reviewer’s keen interest in exploring additional parameter-efficient finetuning methods and identifying trends across datasets, domains and tasks. In future work we intend to include a more detailed section dedicated to evaluating various parameter-efficient finetuning approaches beyond LoRA and Adapters, as well as the combination thereof. We intend to provide a comparative analysis of these methods across diverse scenarios and extending our work with more challenging tasks. Our goal is to build up on this work and continue building an in-depth examination of trends emerging from the conducted experiments on a wider range of datasets and tasks.

---

### Meta-Review · Area_Chair_baZH · 2023-09-07

**Recommendation:** 2

**Metareview:**

The main concern raised by the reviewers is the lack of novelty. While Adapter and LoRA results for the financial domain are new and provide a data point for PEFT vs full-finetuning, the paper does not contain new ideas.

The soundness of the paper is mixed: some reviewers mention some specific problems in the methodology, and the authors highlight that they do a full grid search to find the right hyperparameter, thus providing a strong methodology. Some other issues remain partially unaddressed such as the reproduction of a core result. All in all, this is a borderline paper that could benefit from further revision and review. In this state, I would only recommend acceptance to findings if there is there are enough slots in the proceeding.

---

### Decision · Program_Chairs · 2023-10-07

**Decision:**

Accept-Findings

**Comment:**

The main concern raised by the reviewers is the lack of novelty. While Adapter and LoRA results for the financial domain are new and provide a data point for PEFT vs full-finetuning, the paper does not contain new ideas.

The soundness of the paper is mixed: some reviewers mention some specific problems in the methodology, and the authors highlight that they do a full grid search to find the right hyperparameter, thus providing a strong methodology. Some other issues remain partially unaddressed such as the reproduction of a core result. All in all, this is a borderline paper that could benefit from further revision and review. In this state, I would only recommend acceptance to findings if there is there are enough slots in the proceeding.